# Sparse PCA with Oracle Property

**Quanquan Gu**
Department of Operations Research
and Financial Engineering
Princeton University
Princeton, NJ 08544, USA
qgu@princeton.edu

**Zhaoran Wang**
Department of Operations Research
and Financial Engineering
Princeton University
Princeton, NJ 08544, USA
zhaoran@princeton.edu

**Han Liu**
Department of Operations Research
and Financial Engineering
Princeton University
Princeton, NJ 08544, USA
hanliu@princeton.edu

## Abstract

In this paper, we study the estimation of the $k$-dimensional sparse principal subspace of covariance matrix $\Sigma$ in the high-dimensional setting. We aim to recover the oracle principal subspace solution, i.e., the principal subspace estimator obtained assuming the true support is known a priori. To this end, we propose a family of estimators based on the semidefinite relaxation of sparse PCA with novel regularizations. In particular, under a weak assumption on the magnitude of the population projection matrix, one estimator within this family exactly recovers the true support with high probability, has exact rank-$k$, and attains a $\sqrt{s/n}$ statistical rate of convergence with $s$ being the subspace sparsity level and $n$ the sample size. Compared to existing support recovery results for sparse PCA, our approach does not hinge on the spiked covariance model or the limited correlation condition. As a complement to the first estimator that enjoys the oracle property, we prove that, another estimator within the family achieves a sharper statistical rate of convergence than the standard semidefinite relaxation of sparse PCA, even when the previous assumption on the magnitude of the projection matrix is violated. We validate the theoretical results by numerical experiments on synthetic datasets.

## 1 Introduction

Principal Component Analysis (PCA) aims at recovering the top $k$ leading eigenvectors $\boldsymbol{u}_1, \ldots, \boldsymbol{u}_k$ of the covariance matrix $\Sigma$ from sample covariance matrix $\widehat{\Sigma}$. In applications where the dimension $p$ is much larger than the sample size $n$, classical PCA could be inconsistent [12]. To avoid this problem, one common assumption is that the leading eigenvector $\boldsymbol{u}_1$ of the population covariance matrix $\Sigma$ is sparse, i.e., the number of nonzero elements in $\boldsymbol{u}_1$ is less than the sample size, $s = \text{supp}(\text{u}_1) < \text{n}$. This gives rise to Sparse Principal Component Analysis (SPCA). In the past decade, significant progress has been made toward the methodological development [13, 8, 30, 22, 7, 14, 28, 19, 27] as well as theoretical understanding [12, 20, 1, 24, 21, 4, 6, 3, 2, 18, 15] of sparse PCA.

However, all the above studies focused on estimating the leading eigenvector $\boldsymbol{u}_1$. When the top $k$ eigenvalues of $\Sigma$ are not distinct, there exist multiple groups of leading eigenvectors that are equivalent up to rotation. In order to address this problem, it is reasonable to de-emphasize eigenvectors and to instead focus on their span $\mathcal{U}$, i.e., the principal subspace of variation. [23, 25, 16, 27]

proposed *Subspace Sparsity*, which defines sparsity on the projection matrix onto subspace $\mathcal{U}$, i.e., $\Pi^* = UU^\top$, as the number of nonzero entries on the diagonal of $\Pi^*$, i.e., $s = |\text{supp}(\text{diag}(\Pi^*))|$. They proposed to estimate the principal subspace instead of principal eigenvectors of $\Sigma$, based $\ell_{1,1}$-norm regularization on a convex set called Fantope [9], that provides a tight relaxation for simultaneous rank and orthogonality constraints on the positive semidefinite cone. The convergence rate of their estimator is $O(\lambda_1/(\lambda_k - \lambda_{k+1})s\sqrt{\log p/n})$, where $\lambda_k, k = 1, \ldots, p$ is the $k$-th largest eigenvalue of $\Sigma$. Moreover, their support recovery relies on limited correlation condition (LCC) [16], which is similar to irrepresentable condition in sparse linear regression. We notice that [1] also analyzed the semidefinite relaxation of sparse PCA. However, they only considered rank-1 principal subspace and the stringent spiked covariance model, where the population covariance matrix is block diagonal.

In this paper, we aim to recover the oracle principal subspace solution, i.e., the principal subspace estimator obtained assuming the true support is known a priori. Based on recent progress made on penalized $M$-estimators with nonconvex penalty functions [17, 26], we propose a family of estimators based on the semidefinite relaxation of sparse PCA with novel regularizations. It estimates the $k$-dimensional principal subspace of a population matrix $\Sigma$ based on its empirical version $\widehat{\Sigma}$. In particular, under a weak assumption on the magnitude of the projection matrix, i.e,

$$\min_{(i,j) \in T} |\Pi_{ij}^*| \geq \nu + \frac{C\sqrt{k}\lambda_1}{\lambda_k - \lambda_{k+1}}\sqrt{\frac{s}{n}},$$

where $T$ is the support of $\Pi^*$, $\nu$ is a parameter from nonconvex penalty and $C$ is an universal constant, one estimator within this family exactly recovers the oracle solution with high probability, and is exactly of rank $k$. It is worth noting that unlike the linear regression setting, where the estimators that can recover the oracle solution often have nonconvex formulations, our estimator here is obtained from a convex optimization[1], and has unique global solution. Compared to existing support recovery results for sparse PCA, our approach does not hinge on the spiked covariance model [1] or the limited correlation condition [16]. Moreover, it attains the same convergence rate as standard PCA as if the support of the true projection matrix is provided a priori. More specifically, the Frobenius norm error of the estimator $\widehat{\Pi}$ is bounded with high probability as follows

$$\|\widehat{\Pi} - \Pi^*\|_F \leq \frac{C\lambda_1}{\lambda_k - \lambda_{k+1}}\sqrt{\frac{ks}{n}},$$

where $k$ is the dimension of the subspace.

As a complement to the first estimator that enjoys the oracle property, we prove that, another estimator within the family achieves a sharper statistical rate of convergence than the standard semidefinite relaxation of sparse PCA, even when the previous assumption on the magnitude of the projection matrix is violated. This estimator is based on nonconvex optimizaiton. With a suitable choice of the regularization parameter, we show that any local optima to the optimization problem is a good estimator for the projection matrix of the true principal subspace. In particular, we show that the Frobenius norm error of the estimator $\widehat{\Pi}$ is bounded with high probability as

$$\|\widehat{\Pi} - \Pi^*\|_F \leq \frac{C\lambda_1}{\lambda_k - \lambda_{k+1}}\left(\sqrt{\frac{s_1 s}{n}} + \sqrt{m_1 m_2 \frac{\log p}{n}}\right),$$

where $s_1, m_1, m_2$ are all no larger than $s$. Evidently, it is sharper than the convergence rate proved in [23]. Note that the above rate consists of two terms, the $O(\sqrt{s_1 s/n})$ term corresponds to the entries of projection matrix satisfying the previous assumption (i.e., with large magnitude), while $O(\sqrt{m_1 m_2 \log p/n})$ corresponds to the entries of projection matrix violating the previous assumption (i.e., with small magnitude).

Finally, we demonstrate the numerical experiments on synthetic datasets, which support our theoretical analysis.

The rest of this paper is arranged as follows. Section 2 introduces two estimators for the principal subspace of a covariance matrix. Section 3 analyzes the statistical properties of the two estimators.

We present an algorithm for solving the estimators in Section 4. Section 5 shows the experiments on synthetic datasets. Section 6 concludes this work with remarks.

**Notation.** Let $[p]$ be the shorthand for $\{1, \ldots, p\}$. For matrices $A$, $B$ of compatible dimension, $\langle A, B \rangle := \text{tr}(A^\top B)$ is the Frobenius inner product, and $\|A\|_F = \langle A, A \rangle$ is the squared Frobenius norm. $\|x\|_q$ is the usual $\ell_q$ norm with $\|x\|_0$ defined as the number of nonzero entries of $x$. $\|A\|_{a,b}$ is the $(a,b)$-norm defined to be the $\ell_b$ norm of the vector of rowwise $\ell_a$ norms of $A$, e.g. $\|A\|_{1,\infty}$ is the maximum absolute row sum. $\|A\|_2$ is the spectral norm of $A$, and $\|A\|_*$ is the trace norm (nuclear norm) of $A$. For a symmetric matrix $A$, we define $\lambda_1(A) \geq \lambda_2(A) \geq \ldots \geq \lambda_p(A)$ to be the eigenvalues of $A$ with multiplicity. When the context is obvious we write $\lambda_j = \lambda_j(A)$ as shorthand.

## 2 The Proposed Estimators

In this section, we present a family of estimators based on the semidefinite relaxation of sparse PCA with novel regularizations, for the principal subspace of the population covariance matrix. Before going into the details of the proposed estimators, we first present the formal definition of principal subspace estimation.

### 2.1 Problem Definition

Let $\Sigma \in \mathbb{R}^{p \times p}$ be an unknown covariance matrix, with eigen-decomposition as follows

$$\Sigma = \sum_{i=1}^{p} \lambda_i \mathbf{u}_i \mathbf{u}_i^\top,$$

where $\lambda_1 \geq \ldots \geq \lambda_p$ are eigenvalues (with multiplicity) and $\mathbf{u}_1, \ldots, \mathbf{u}_p \in \mathbb{R}^p$ are the associated eigenvectors. The $k$-dimensional principal subspace of $\Sigma$ is the subspace spanned by $\mathbf{u}_1, \ldots, \mathbf{u}_k$. The projection matrix to the $k$-dimensional principal subspace is

$$\Pi^* = \sum_{i=1}^{k} \mathbf{u}_i \mathbf{u}_i^\top = UU^\top,$$

where $U = [\mathbf{u}_1, \ldots, \mathbf{u}_k]$ is an orthonormal matrix. The reason why principal subspace is more appealing is that it avoids the problem of un-identifiability of eigenvectors when the eigenvalues are not distinct. In fact, we only need to assume $\lambda_k - \lambda_{k+1} > 0$ instead of $\lambda_1 > \ldots > \lambda_k > \lambda_{k+1}$. Then the principal subspace $\Pi^*$ is unique and identifiable. We also assume that $k$ is fixed.

Next, we introduce the definition of *Subspace Sparsity* [25], which can be seen as the extension of conventional *Eigenvector Sparsity* used in sparse PCA.

**Definition 1.** *[25] (Subspace Sparsity) The projection $\Pi^*$ onto the subspace spanned by the eigenvectors of $\Sigma$ corresponding to its $k$ largest eigenvalues satisfies $\|U\|_{2,0} \leq s$, or equivalently $\|\text{diag}(\Pi)\|_0 \leq s$.*

In the extreme case that $k = 1$, the support of the projection matrix onto the rank-1 principal subspace is the same as the support of the sparse leading eigenvector.

The problem definition of principal subspace estimation is: given an i.i.d. sample $\{\mathbf{x}_1, \mathbf{x}_2, \ldots, \mathbf{x}_n\} \subset \mathbb{R}^p$ which are drawn from an unknown distribution of zero mean and covariance matrix $\Sigma$, we aim to estimate $\Pi^*$ based on the empirical covariance matrix $S \in \mathbb{R}^{p \times p}$, that is given by $\widehat{\Sigma} = 1/n \sum_{i=1}^{n} \mathbf{x}_i \mathbf{x}_i^\top$. We are particularly interested in the high dimensional setting, where $p \to \infty$ as $n \to \infty$, in sharp contrast to conventional setting where $p$ is fixed and $n \to \infty$.

Now we are ready to design a family of estimators for $\Pi^*$.

### 2.2 A Family of Sparse PCA Estimators

Given a sample covariance matrix $\widehat{\Sigma} \in \mathbb{R}^{p \times p}$, we propose a family of sparse principal subspace estimator $\widehat{\Pi}$ that is defined to be a solution of the semidefinite relaxation of sparse PCA

$$\widehat{\Pi}_\tau = \underset{\Pi}{\text{argmin}} \; -\langle \widehat{\Sigma}, \Pi \rangle + \frac{\tau}{2} \|\Pi\|_F^2 + \mathcal{P}_\lambda(\Pi), \quad \text{subject to } \Pi \in \mathcal{F}^k, \tag{1}$$

where $\tau > 0, \lambda > 0$ is a regularization parameter, $\mathcal{F}^k$ is a convex body called the Fantope [9, 23], that is defined as follows

$$\mathcal{F}^k = \{X : 0 \prec X \prec I \text{ and } \operatorname{tr}(X) = k\},$$

and $\mathcal{P}_\lambda(\Pi)$ is a decomposable nonconvex penalty, i.e., $\mathcal{P}_\lambda(\Pi) = \sum_{i,j=1}^{p} p_\lambda(\Pi_{ij})$. Typical nonconvex penalties include the smoothly clipped absolute deviation (SCAD) penalty [10] and minimax concave penalty MCP [29], which can eliminate the estimation bias and attain more refined statistical rates of convergence [17, 26]. For example, MCP penalty is defined as

$$p_\lambda(t) = \lambda \int_0^{|t|} \left(1 - \frac{z}{\lambda b}\right) dz = \left(\lambda|t| - \frac{t^2}{2b}\right) \mathbf{1}(|t| \le b\lambda) + \frac{b\lambda^2}{2} \mathbf{1}(|t| > b\lambda), \tag{2}$$

where $b > 0$ is a fixed parameter.

An important property of the nonconvex penalties $p_\lambda(t)$ is that they can be formulated as the sum of the $\ell_1$ penalty and a concave part $q_\lambda(t)$: $p_\lambda(t) = \lambda|t| + q_\lambda(t)$. For example, if $p_\lambda(t)$ is chosen to be the MCP penalty, then the corresponding $q_\lambda(t)$ is:

$$q_\lambda(t) = -\frac{t^2}{2b}\mathbf{1}(|t| \le b\lambda) + \left(\frac{b\lambda^2}{2} - \lambda|t|\right)\mathbf{1}(|t| > b\lambda),$$

We rely on the following regularity conditions on $p_\lambda(t)$ and its concave component $q_\lambda(t)$:

(a) $p_\lambda(t)$ satisfies $p'_\lambda(t) = 0$, for $|t| \ge \nu > 0$.

(b) $q'_\lambda(t)$ is monotone and Lipschitz continuous, i.e., for $t' \ge t$, there exists a constant $\zeta_- \ge 0$ such that

$$-\zeta_- \le \frac{q'_\lambda(t') - q'_\lambda(t)}{t' - t}.$$

(c) $q_\lambda(t)$ and $q'_\lambda(t)$ pass through the origin, i.e., $q_\lambda(0) = q'_\lambda(0) = 0$.

(d) $q'_\lambda(t)$ is bounded, i.e., $|q'_\lambda(t)| \le \lambda$ for any $t$.

The above conditions apply to a variety of nonconvex penalty functions. For example, for MCP in (2), we have $\nu = b\lambda$ and $\zeta_- = 1/b$.

It is easy to show that when $\tau > \zeta_-$, the problem in (1) is strongly convex, and therefore its solution is unique. We notice that [16] also introduced the same regularization term $\tau/2\|\Pi\|_F^2$ in their estimator. However, our motivation is quite different from theirs. We introduce this term because it is essential for the estimator in (1) to achieve the oracle property provided that the magnitude of all the entries in the population projection matrix is sufficiently large. We call (1) *Convex Sparse PCA Estimator*.

Note that constraint $\Pi \in \mathcal{F}^k$ only guarantees that the rank of $\widehat{\Pi}$ is $\ge k$. However, we can prove that our estimator is of rank $k$ exactly. This is in contrast to [23], where some post projection is needed, to make sure their estimator is of rank $k$.

## 2.3 Nonconvex Sparse PCA Estimator

In the case that the magnitude of entries in the population projection matrix $\Pi^*$ violates the previous assumption, (1) with $\tau > \zeta_-$ no longer enjoys the desired oracle property. To this end, we consider another estimator from the family of estimators in (1) with $\tau = 0$,

$$\widehat{\Pi}_{\tau=0} = \underset{\Pi}{\operatorname{argmin}} -\langle\widehat{\Sigma}, \Pi\rangle + \mathcal{P}_\lambda(\Pi), \quad \text{subject to } \Pi \in \mathcal{F}^k. \tag{3}$$

Since $-\langle\widehat{\Sigma}, \Pi\rangle$ is an affine function, and $\mathcal{P}_\lambda(\Pi)$ is nonconvex, the estimator in (3) is nonconvex. We simply refer to it as *Nonconvex Sparse PCA Estimator*. We will prove that it achieves a sharper statistical rate of convergence than the standard semidefinite relaxation of sparse PCA [23], even when the previous assumption on the magnitude of the projection matrix is violated.

It is worth noting that although our estimators in (1) and (3) are for the projection matrix $\Pi$ of the principal subspace, we can also provide an estimator of $U$. By definition, the true subspace satisfies

$\Pi^* = UU^\top$. Thus, the estimator $\widehat{U}$ can be computed from $\widehat{\Pi}$ using eigenvalue decomposition. In detail, we can set the columns of $\widehat{U}$ to be the top k leading eigenvectors of $\widehat{\Pi}$. In case that the top k eigenvalues of $\widehat{\Pi}$ are the same, we can follow the standard PCA convention by rotating the eigenvectors with a rotation matrix $R$, such that $(\widehat{U}R)^T \widehat{\Sigma}(\widehat{U}R)$ is diagonal. Then $\widehat{U}R$ is the orthonormal basis for the estimated principal subspace, and can be used for visualization and dimension reduction.

## 3 Statistical Properties of the Proposed Estimators

In this section, we present the statistical properties of the two estimators in the family (1). One is with $\tau > \zeta_-$, the other is with $\tau = 0$. The proofs are all included in the longer version of this paper.

To evaluate the statistical performance of the principal subspace estimators, we need to define the estimator error between the estimated projection matrix and the true projection matrix. In our study, we use the Frobenius norm error $\|\widehat{\Pi} - \Pi^*\|_F$.

### 3.1 Oracle Property and Convergence Rate of Convex Sparse PCA

We first analyze the estimator in (1) when $\tau > \zeta_-$. We prove that, the estimator $\widehat{\Pi}$ in (1) recovers the support of $\Pi^*$ under suitable conditions on its magnitude. Before we present this theorem, we introduce the definition of an oracle estimator, denoted by $\widehat{\Pi}_O$. Recall that $S = \text{supp}(\text{diag}(\Pi^*))$. The oracle estimator $\widehat{\Pi}_O$ is defined as

$$\widehat{\Pi}_O = \underset{\text{supp}(\text{diag}(\Pi))\subset S, \Pi\in\mathcal{F}^k}{\text{argmin}} \mathcal{L}(\Pi). \tag{4}$$

where $\mathcal{L}(\Pi) = -\langle\widehat{\Sigma},\Pi\rangle + \frac{\tau}{2}\|\Pi\|_F^2$. Note that the above oracle estimator is not a practical estimator, because we do not know the true support $S$ in practice.

The following theorem shows that, under suitable conditions, $\widehat{\Pi}$ in (1) is the same as the oracle estimator $\widehat{\Pi}_O$ with high probability, and therefore exactly recovers the support of $\Pi^*$.

**Theorem 1.** *(Support Recovery) Suppose the nonconvex penalty* $\mathcal{P}_\lambda(\Pi) = \sum_{i,j=1}^p p_\lambda(\Pi)$ *satisfies conditions (a) and (b). If* $\Pi^*$ *satisfies* $\min_{(i,j)\in T}|\Pi^*_{ij}| \geq \nu + C\sqrt{k}\lambda_1/(\lambda_k - \lambda_{k+1})\sqrt{s/n}$. *For the estimator in* (1) *with the regularization parameter* $\lambda = C\lambda_1\sqrt{\log p/n}$ *and* $\tau > \zeta_-$*, we have with probability at least* $1 - 1/n^2$ *that* $\widehat{\Pi} = \widehat{\Pi}_O$*, which further implies* $\text{supp}(\text{diag}(\widehat{\Pi})) = \text{supp}(\text{diag}(\widehat{\Pi}_O)) = \text{supp}(\text{diag}(\Pi^*))$ *and* $\text{rank}(\widehat{\Pi}) = \text{rank}(\widehat{\Pi}_O) = k$.

For example, if we use MCP penalty, the magnitude assumption turns out to be $\min_{(i,j)\in T}|\Pi^*_{ij}| \geq Cb\lambda_1\sqrt{\log p/n} + C\sqrt{k}\lambda_1/(\lambda_k - \lambda_{k+1})\sqrt{s/n}$.

Note that in our proposed estimator in (1), we do not rely on any oracle knowledge on the true support. Our theory in Theorem 1 shows that, with high probability, the estimator is identical to the oracle estimator, and thus exactly recovers the true support.

Compared to existing support recovery results for sparse PCA [1, 16], our condition on the magnitude is weaker. Note that the limited correlation condition [16] and the even stronger spiked covariance condition [1] impose constraints not only on the principal subspace corresponding to $\lambda_1, \ldots, \lambda_k$, but also on the "non-signal" part, i.e., the subspace corresponding to $\lambda_{k+1}, \ldots, \lambda_p$. Unlike these conditions, we only impose conditions on the "signal" part, i.e., the magnitude of the projection matrix $\Pi^*$ corresponding to to $\lambda_1, \ldots, \lambda_k$. We attribute the oracle property of our estimator to novel regularizations ($\tau/2\|\Pi\|_F^2$ plus nonconvex penalty).

The oracle property immediately implies that under the above conditions on the magnitude, the estimator in (1) achieves the convergence rate of standard PCA as if we know the true support $S$ a priori. This is summarized in the following theorem.

**Theorem 2.** *Under the same conditions of Theorem 1, we have with probability at least* $1 - 1/n^2$ *that*

$$\|\widehat{\Pi} - \Pi^*\|_F \leq \frac{C\sqrt{k}\lambda_1}{\lambda_k - \lambda_{k+1}}\sqrt{\frac{s}{n}},$$

*for some universal constant $C$.*

Evidently, the estimator attains a much sharper statistical rate of convergence than the state-of-the-art result proved in [23].

## 3.2 Convergence Rate of Nonconvex Sparse PCA

We now analyze the estimator in (3), which is a special case of (1) when $\tau = 0$. We basically show that any local optima of the non-convex optimization problem in (3) is a good estimator. In other words, our theory applies to any projection matrix $\widehat{\Pi}_{\tau=0} \in \mathbb{R}^{p \times p}$ that satisfies the first-order necessary conditions (variational inequality) to be a local minimum of (3):

$$\langle \widehat{\Pi}_{\tau=0} - \Pi', -\widehat{\Sigma} + \nabla \mathcal{P}_\lambda(\widehat{\Pi}) \rangle \leq 0, \ \forall \, \Pi' \in \mathcal{F}^k$$

Recall that $S = \text{supp}(\text{diag}(\Pi^*))$ with $|S| = s$, $T = S \times S$ with $|T| = s^2$, and $T^c = [p] \times [p] \setminus T$. For $(i, j) \in T_1 \subset T$ with $|T_1| = t_1$, we assume $|\Pi_{ij}^*| \geq \nu$, while for $(i, j) \in T_2 \subset T$ with $|T_2| = t_2$, we assume $|\Pi_{ij}^*| < \nu$. Clearly, we have $s^2 = t_1 + t_2$. There exists a minimal submatrix $A \in \mathbb{R}^{n_1 \times n_2}$ of $\Pi^*$, which contains all the elements in $T_1$, with $s_1 = \min\{n_1, n_2\}$. There also exists a minimal submatrix $B \in \mathbb{R}^{m_1 \times m_2}$ of $\Pi^*$, that contains all the elements in $T_2$.

Note that in general, $s_1 \leq s$, $m_1 \leq s$ and $m_2 \leq s$. In the worst case, we have $s_1 = m_1 = m_2 = s$.

**Theorem 3.** *Suppose the nonconvex penalty $\mathcal{P}_\lambda(\Pi) = \sum_{i,j=1}^p p_\lambda(\Pi)$ satisfies conditions (b) (c) and (d). For the estimator in* (3) *with regularization parameter $\lambda = C\lambda_1\sqrt{\log p / n}$ and $\zeta_- \leq (\lambda_k - \lambda_{k+1})/4$, with probability at least $1 - 4/p^2$, any local optimal solution $\widehat{\Pi}_{\tau=0}$ satisfies*

$$\|\widehat{\Pi}_{\tau=0} - \Pi^*\|_F \leq \underbrace{\frac{4C\lambda_1\sqrt{s_1}}{(\lambda_k - \lambda_{k+1})}\sqrt{\frac{s}{n}}}_{T_1 : |\Pi_{ij}^*| \geq \nu} + \underbrace{\frac{12C\lambda_1\sqrt{m_1 m_2}}{(\lambda_k - \lambda_{k+1})}\sqrt{\frac{\log p}{n}}}_{T_2 : |\Pi_{ij}^*| < \nu} \, .$$

Note that the upper bound can be decomposed into two parts according to the magnitude of the entries in the true projection matrix, i.e., $|\Pi_{ij}^*|, 1 \leq i, j \leq p$. We have the following comments:

On the one hand, for those strong "signals", i.e., $|\Pi_{ij}^*| \geq \nu$, we are able to achieve the convergence rate of $O(\lambda_1\sqrt{s_1}/(\lambda_k - \lambda_{k+1})\sqrt{s/n})$. Since $s_1$ is at most equal to $s$, the worst-case rate is $O(\lambda_1/(\lambda_k - \lambda_{k+1})s/\sqrt{n})$, which is sharper than the rate proved in [23], i.e., $O(\lambda_1/(\lambda_k - \lambda_{k+1})s\sqrt{\log p / n})$. In the other case that $s_1 < s$, the convergence rate could be even sharper.

On the other hand, for those weak "signals", i.e., $|\Pi_{ij}^*| < \nu$, we are able to achieve the convergence rate of $O(\lambda_1\sqrt{m_1 m_2}/(\lambda_k - \lambda_{k+1})\sqrt{\log p / n})$. Since both $m_1$ and $m_2$ are at most equal to $s$, the worst-case rate is $O(\lambda_1/(\lambda_k - \lambda_{k+1})s\sqrt{\log p / n})$, which is the same as the rate proved in [23]. In the other case that $\sqrt{m_1 m_2} < s$, the convergence rate will be sharper than that in [23].

The above discussions clearly demonstrate the advantage of our estimator, which essentially benefits from non-convex penalty.

## 4 Optimization Algorithm

In this section, we present an optimization algorithm to solve (1) and (3). Since (3) is a special case of (1) with $\tau = 0$, it is sufficient to develop an algorithm for solving (1).

Observing that (1) has both nonsmooth regularization term and nontrivial constraint set $\mathcal{F}^k$, it is difficult to directly apply gradient descent and its variants. Following [23], we present an alternating direction method of multipliers (ADMM) algorithm. The proposed ADMM algorithm can efficiently compute the global optimum of (1). It can also find a local optimum to (3). It is worth noting that other algorithms such as Peaceman Rachford Splitting Method [11] can also be used to solve (1).

We introduce an auxiliary variable $\Phi \in \mathbb{R}^{p \times p}$, and consider an equivalent form of (1) as follows

$$\underset{\Pi, \Phi}{\text{argmin}} -\langle \widehat{\Sigma}, \Pi \rangle + \frac{\tau}{2}\|\Pi\|_F^2 + \mathcal{P}_\lambda(\Phi), \quad \text{subject to } \Pi = \Phi, \ \Pi \in \mathcal{F}^k. \tag{5}$$

The augmented Lagrangian function corresponding to (5) is

$$L(\Pi, \Phi, \Theta) = \infty \mathbf{1}_{\mathcal{F}^k}(\Pi) - \langle \widehat{\Sigma}, \Pi \rangle + \frac{\tau}{2} \|\Pi\|_F^2 + \mathcal{P}_\lambda(\Phi) + \langle \Theta, \Pi - \Phi \rangle + \frac{\rho}{2} \|\Pi - \Phi\|_F^2, \quad (6)$$

where $\Theta \in \mathbb{R}^{d \times d}$ is the Lagrange multiplier associated with the equality constraint $\Pi = \Phi$ in (5), and $\rho > 0$ is a penalty parameter that enforces the equality constraint $\Pi = \Phi$. The detailed update scheme is described in Algorithm 1. In details, the first subproblem (Line 5 of Algorithm 1) can be solved by projecting $\rho/(\rho+\tau)\Phi^{(t)} - 1/(\rho+\tau)\Theta^{(t)} + 1/(\rho+\tau)\widehat{\Sigma}$ onto Fantope $\mathcal{F}^k$. This projection has a simple form solution as shown by [23, 16]. The second subproblem (Line 6 of Algorithm 1) can be solved by generalized soft-thresholding operator as shown by [5] [17].

---

**Algorithm 1** Solving Convex Relaxation (5) using ADMM.

---

1: **Input:** Covariance Matrix Estimator $\widehat{\Sigma}$
2: **Parameter:** Regularization parameters $\lambda > 0, \tau \geq 0$, Penalty parameter $\rho > 0$ of the augmented Lagrangian, Maximum number of iterations $T$
3: $\Pi^{(0)} \leftarrow \mathbf{0}, \Phi^{(0)} \leftarrow \mathbf{0}, \Theta^{(0)} \leftarrow \mathbf{0}$
4: **For** $t = 0, \ldots, T-1$
5: $\qquad \Pi^{(t+1)} \leftarrow \arg\min_{\Pi \in \mathcal{F}^k} \frac{1}{2} \|\Pi - (\frac{\rho}{\rho+\tau}\Phi^{(t)} - \frac{1}{\rho+\tau}\Theta^{(t)} + \frac{1}{\rho+\tau}\widehat{\Sigma})\|_F^2$
6: $\qquad \Phi^{(t+1)} \leftarrow \arg\min_\Phi \frac{1}{2} \|\Phi - (\Pi^{(t+1)} + \frac{1}{\rho}\Theta^{(t)})\|_F^2 + \mathcal{P}_{\frac{\lambda}{\rho}}(\Phi)$
7: $\qquad \Theta^{(t+1)} \leftarrow \Theta^{(t)} + \rho(\Pi^{(t+1)} - \Phi^{(t+1)})$
8: **End For**
9: **Output:** $\Pi^{(T)}$

---

## 5 Experiments

In this section, we conduct simulations on synthetic datasets to validate the effectiveness of the proposed estimators in Section 2. We generate two synthetic datasets via designing two covariance matrices. The covariance matrix $\Sigma$ is basically constructed through the eigenvalue decomposition. In detail, for synthetic dataset I, we set $s = 5$ and $k = 1$. The leading eigenvalue of its covariance matrix $\Sigma$ is set as $\lambda_1 = 100$, and its corresponding eigenvector is sparse in the sense that only the first $s = 5$ entries are nonzero and set be to $1/\sqrt{5}$. The other eigenvalues are set as $\lambda_2 = \ldots = \lambda_p = 1$, and their eigenvectors are chosen arbitrarily. For synthetic dataset II, we set $s = 10$ and $k = 5$. The top-5 eigenvalues are set as $\lambda_1 = \ldots = \lambda_4 = 100$ and $\lambda_5 = 10$. We generate their corresponding eigenvectors by sampling its nonzero entries from a standard Gaussian distribution, and then orthnormalizing them while retaining the first $s = 10$ rows nonzero. The other eigenvalues are set as $\lambda_6 = \ldots = \lambda_p = 1$, and the associated eigenvectors are chosen arbitrarily. Based on the covariance matrix, the groundtruth rank-$k$ projection matrix $\Pi^*$ can be immediately calculated. Note that synthetic dataset II is more challenging than synthetic dataset I, because the smallest magnitude of $\Pi^*$ in synthetic dataset I is 0.2, while that in synthetic dataset II is much smaller (about $10^{-3}$). We sample $n = 80$ i.i.d. observations from a normal distribution $\mathcal{N}(0, \Sigma)$ with $p = 128$, and then calculate the sample covariance matrix $\widehat{\Sigma}$.

Since the focus of this paper is principal subspace estimation rather than principal eigenvectors estimation, it is sufficient to compare our proposed estimators (*Convex SPCA* in (1) and *Nonconvex SPCA* in 3) with the estimator proposed in [23], which is referred to as *Fantope SPCA*. Note that *Fantope PCA* is the pioneering and the state-of-the-art estimator for principal subspace estimation of SPCA. However, since Fantope SPCA uses convex penalty $\|\Pi\|_{1,1}$ on the projection matrix $\Pi$, the estimator is biased [29]. We also compare our proposed estimators with the oracle estimator in (4), which is not a practical estimator but provides the optimal results that we could achieve. In our experiments, we need to compare the estimator attained by the algorithmic procedure and the oracle estimator. To obtain the oracle estimator, we apply standard PCA on the submatrix (supported on the true support) of the sample covariance $\widehat{\Sigma}$. Note that the true support is known because we use synthetic datasets here.

In order to evaluate the performance of the above estimators, we look at the Frobenius norm error $\|\widehat{\Pi} - \Pi^*\|_F$. We also use True Positive Rate (TPR) and False Positive Rate (FPR) to evaluate the

support recovery result. The larger the TPR and the smaller the FPR, the better the support recovery result.

Both of our estimators use MCP penalty, though other nonconvex penalties such as SCAD could be used as well. In particular, we set $b = 3$. For *Convex SPCA*, we set $\tau = \frac{2}{b}$. The regularization parameter $\lambda$ in our estimators as well as *Fantope SPCA* is tuned by 5-fold cross validation on a held-out dataset. The experiments are repeated 20 times, and the mean as well as the standard errors are reported. The empirical results on synthetic datasets I and II are displayed in Table 1.

Table 1: Empirical results for subspace estimation on synthetic datasets I and II.

| Synthetic I | | $\|\widehat{\Pi} - \Pi^*\|_F$ | TPR | FPR |
|---|---|---|---|---|
| $n = 80$ | Oracle | 0.0289±0.0134 | 1 | 0 |
| $p = 128$ | Fantope SPCA | 0.0317±0.0149 | 1.0000±0.0000 | 0.0146±0.0218 |
| $s = 5$ | Convex SPCA | 0.0290±0.0132 | 1.0000±0.0000 | 0.0000±0.0000 |
| $k = 1$ | Nonconvex SPCA | 0.0290±0.0133 | 1.0000±0.0000 | 0.0000±0.0000 |

| Synthetic II | | $\|\widehat{\Pi} - \Pi^*\|_F$ | TPR | FPR |
|---|---|---|---|---|
| $n = 80$ | Oracle | 0.1487±0.0208 | 1 | 0 |
| $p = 128$ | Fantope SPCA | 0.2788±0.0437 | 1.0000±0.0000 | 0.8695±0.1634 |
| $s = 10$ | Convex SPCA | 0.2031±0.0331 | 1.0000±0.0000 | 0.5814±0.0674 |
| $k = 5$ | Nonconvex SPCA | 0.2041±0.0326 | 1.0000±0.0000 | 0.6000±0.0829 |

It can be observed that both *Convex SPCA* and *Nonconvex SPCA* estimators outperform *Fantope SPCA* estimator [23] greatly in both datasets. In details, on synthetic dataset I with relatively large magnitude of $\Pi^*$, our *Convex SPCA* estimator achieves the same estimation error and perfect support recovery as the oracle estimator. This is consistent with our theoretical results in Theorems 1 and 2. In addition, our *Nonconvex SPCA* estimator achieves very similar results with *Convex SPCA*. This is not very surprising, because provided that the magnitude of all the entries in $\Pi^*$ is large, *Nonconvex SPCA* attains a rate which is only $1/\sqrt{s}$ slower than *Convex SPCA*. *Fantope SPCA* cannot recover the support perfectly because it detected several false positive supports. This implies that the LCC condition is stronger than our large magnitude assumption, and does not hold on this dataset.

On synthetic dataset II, our *Convex SPCA* estimator does not perform as well as the oracle estimator. This is because the magnitude of $\Pi^*$ is small (about $10^{-3}$). Given the sample size $n = 80$, the conditions of Theorems 1 are violated. But note that *Convex SPCA* is still slightly better than *Nonconvex SPCA*. And both of them are much better than *Fantope SPCA*. This again illustrates the superiority of our estimators over existing best approach, i.e., *Fantope SPCA* [23].

## 6  Conclusion

In this paper, we study the estimation of the $k$-dimensional principal subspace of a population matrix $\Sigma$ based on sample covariance matrix $\widehat{\Sigma}$. We proposed a family of estimators based on novel regularizations. The first estimator is based on convex optimization, which is suitable for projection matrix with large magnitude entries. It enjoys oracle property and the same convergence rate as standard PCA. The second estimator is based on nonconvex optimization, and it also attains faster rate than existing principal subspace estimator, even when the large magnitude assumption is violated. Numerical experiments on synthetic datasets support our theoretical results.

## Acknowledgement

We would like to thank the anonymous reviewers for their helpful comments. This research is partially supported by the grants NSF IIS1408910, NSF IIS1332109, NIH R01MH102339, NIH R01GM083084, and NIH R01HG06841.

## Footnotes

[1]Even though we use nonconvex penalty, the resulting problem as a whole is still a convex optimization problem, because we add another strongly convex term in the regularization part, i.e., $\tau/2\|\Pi\|_F$.

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
