[Reviews · NeurIPS 2014]

Submitted by Assigned_Reviewer_10

This paper studies the estimation of the k-dimensional principal subspace of a population matrix based on sample covariance matrix. Two estimators based on convex and non-convex optimizations are developed for projection matrix with large or small magnitude entries, respectively. Both these two estimators are shown to enjoy satisfactory theoretical properties and experimental results compared with state-of-the-art estimators.

It would be better to clearly explain what the oracle knowledge used in the proposed algorithm is, and how to set up the oracle estimator comparison experiments. In Eq. (4), the definition of L is missing.

At line 165, the explicit equation of q(t) is not given. Some example q(t) fulfilling the conditions are necessary to be introduced.

Some brief introduction on the principles of Fantope PCA method is necessary in the experimental setup. Besides, the LCC condition mentioned in the experimental analysis has not been explained.
Summary: A nice paper with novel contribution in providing effective principal subspace estimators of favorable statistical properties.

Submitted by Assigned_Reviewer_23

The abstract is an accurate summary of the paper. The paper proposes two new estimators for a sparse principal component subspace. The estimators are based on a novel regularization of a know semidefinite programming relaxation of the rank-constrained principal subspace problem. Consistency-like properties of the estimators are shown with faster rates that previously known (nearly optimal in some regimes). Suitable optimization algorithms are described and tested on synthetic experiments, showing improved accuracy over previous methods. The new estimators also have weaker structural assumptions on the covariance matrix to guarantee convergence. The asymptotic distribution of one of the estimators is derived, making it possible to construct confidence regions.

The results are fairly interesting, the presentation is clear, the results seem quite original. The significance follows from the improved theoretical rates shown as well as the improved performance in synthetic experiments.

Specific comments:
82: optimizaiton
83: optima -> optimum
141: Confusing notation: the empirical covariance matrix is S as well as \hat \Sigma. Later, S denotes the support.
213: What is L?
220: "If" sentence is incomplete.
Summary: The paper presents theoretically and empirically justified improved estimators for sparse principal component analysis. The results are somewhat incremental.

Submitted by Assigned_Reviewer_39

The authors proposed a general frame work for solving sparse PCA through semidefinite relaxation with both convex and non-convex penalty functions. Certain estimators with convex penalties under this frame can achieve the same convergence rate as the standard PCA. Estimators using non-convex penalties can also achieve higher convergence rate than the existing principal subspace estimator. Both convex and non-convex SPCA demonstrated higher recovery rate than Fantope sPCA.

The manuscript is well written. The framework introduced in this manuscript is incremental to the existing literature. A major limitation of the proposed sPCA is the recovery of U matrix. In practice, the U matrix is used for low dimension visualization. It is not clearly how to use sPCA on dimension reduction.

Overall, the paper represents a significant contribution to the sPCA literature.

Summary: A family of estimators for sparse PCA is introduced in this paper. Same convergence rate as the standard PCA can be achieved for convex SPCA. Higher convergence rate can be attained for non-convex penalty functions. In simulation datasets, a better recovery rate is observed compared to Fantope SPCA.
Author Feedback
Author rebuttal: We would like to thank all the reviewers for their helpful and insightful comments.

Response to Reviewer 1

Thank you very much for your valuable comments.

The L in Eq. (4) is $-\langle \hat{\Sigma}, \Pi \rangle$. We will clearly define L in the final version. In theory, our oracle estimator is defined as the sparse principal subspace estimator obtained given priori knowledge of the true support. However, in our proposed algorithms, we do not rely on any oracle knowledge on the true support. Our theory shows that, with high probability, the estimator obtained by the algorithmic procedure is identical to the oracle estimator, and thus exactly recovers the true support.

In our experiments, we need to compare the estimator attained by the algorithmic procedure and the oracle estimator. To obtain the reference oracle estimator in the experiment, we apply standard PCA on the submatrix (supported on the true support) of the sample covariance \hat{\Sigma}. Note that the true support is known because we use synthetic datasets. We will provide more details in the experiment in the final version.

Thank you for your suggestion on giving an example of q(t). In fact, common nonconvex penalties like SCAD or MCP all satisfy these conditions. We will give the expression of q(t) corresponding to MCP penalty in Eq. (2), to make our presentation more complete.

We will add a brief introduction on Fantope PCA method in the experimental setup. The LCC condition mentioned in the experimental analysis is shorthand for limited correlation condition [17], which is analogous to irrepresentable condition in sparse linear regression. We will add the shorthand “LCC” right after “limited correlation condition” in Line 053 and provide more details in the final version. Thank you very much for pointing out this issue.

Response to Reviewer 2

Thank you very much for your helpful comments.

Thanks for pointing out several typos. We will fix them. For example, we will consistently use $\hat {\Sigma} $ to denote the empirical covariance matrix.
L in Eq. (4) is $-\langle \hat{\Sigma}, \Pi \rangle$. We will define and clarify it.

Response to Reviewer 3

Thank you very much for your nice comments on our work.

“A major limitation of the proposed sPCA is the recovery of U matrix. In practice, the U matrix is used for low dimension visualization. It is not clear how to use sPCA on dimension reduction.”

Our estimator is for the projection matrix of the principal subspace. In fact, by definition, the true subspace satisfies $\Pi^* = U*U^T$. Thus, the estimator $\hat{U}$ can be computed from $\hat{\Pi}$ using eigenvalue decomposition. In detail, we can set the columns of $\hat{U}$ to be the top k leading eigenvectors of $\hat{\Pi}$. In case that the top k eigenvalues of $\hat{\Pi}$ are the same, we can follow the standard PCA convention by rotating the eigenvectors with a rotation matrix $R$, such that $(\hat{U} R)^T \hat{\Sigma} (\hat{U} R)$ is diagonal. Then $\hat{U} R$ is the orthonormal basis for the estimated principal subspace, and can be used for visualization and dimension reduction. We will elaborate and emphasize it in the final version.

Finally, we would like to thank again the reviewers for their positive and valuable comments. We will incorporate the minor revision mentioned above in the final version.